# Study on the Hydraulic Characteristics of the Trapezoidal Energy Dissipation Baffle Block-Step Combination Energy Dissipator

Yu Tian, Yongye Li and Xihuan Sun *

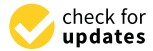



College of Water Resource Science and Engineering, Taiyuan University of Technology, Taiyuan 030024, China; sxjctianyu@163.com (Y.T.); liyongye@tyut.edu.cn (Y.L.)
* Correspondence: sunxihuan@tyut.edu.cn; Tel.: +86-135-1360-0012

**Abstract:** The step-type energy dissipator is widely used to construct small- and medium-sized reservoirs with its high energy dissipation rate. In order to further improve its air entrainment characteristics and energy dissipation, and reduce the influence of cavitation, in this paper, we added a trapezoidal energy dissipation baffle block at the convex corner of the traditional step to form a trapezoidal energy dissipation baffle block-step combination energy dissipator. We used a combination of hydraulic model experiments and numerical simulation to study the hydraulic characteristics. The results showed that the trapezoidal energy dissipation baffle block-step combination energy dissipator initial entrainment point, with the increase in flow rate, gradually moved backward. A step horizontal surface pressure change in the cavity recirculation area showed a prominent "V" shape; in front of the trapezoidal energy dissipation baffle block, there was a rising trend, and in the energy dissipation baffle block gap, there was a declining trend. The step vertical surface pressure showed a decreasing trend, and negative pressure appeared near the convex angle. The cross-section velocity distribution presented a trend of being small at the bottom and large at the surface, with a large velocity gradient in the longitudinal section of the energy dissipation baffle block and a small velocity gradient in the longitudinal section of the nonenergy dissipation baffle block. The energy dissipation rate reached more than 70% within the test range, and the energy dissipation rate gradually decreased with the increase in the flow rate. The combined energy dissipator is conducive to reducing the cavitation hazard and improving the energy dissipation effect, providing a reference for engineering design and existing step energy dissipators to remove risks and reinforcement.

**Keywords:** step dissipator; trapezoidal energy dissipation baffle block; air entrainment characteristics; energy dissipation characteristics

## 1. Introduction

More than 98,000 reservoirs have been built in China, among which more than 94,000 are small reservoirs, accounting for more than 90% of the total [1]. In the construction of small reservoirs, to solve the problem of energy dissipation of the downstream flow, it is imperative to choose an energy dissipation method with good effect and low engineering cost. The step-type energy dissipator is one of the better forms.

The step-type energy dissipator changes the water flow structure, generates an energy dissipation vortex at the step, and promotes the energy dissipation of the water flow through the internal turbulence of the water flow, which effectively increases the energy dissipation effect. Researchers have conducted a series of studies on the hydraulic characteristics of traditional step dissipators. In terms of the water flow pattern, Chanson [2] and Ohtsu et al. [3] proposed the boundary equations of the step nappe flow and skimming flow regime. In terms of step pressure characteristics, Mator et al. [4], Sanchez-Juny et al. [5], and Amador et al. [6] analyzed the pressure variation law on the horizontal and vertical surfaces of the steps, and found that negative pressure is generated on the vertical surface of the step

near the convex corner position, and this area of negative pressure is prone to producing cavitation hazards, which was verified in the experiments of Frizell et al. [7]. In actual projects, cavitation has also occurred to varying degrees in step spillways, such as the New Croton Dam in the United States, the Dona Francisca Dam in Brazil [8], the Danjiangkou Hydropower Station [9], and the step spillway in the Suofengying Hydropower Station [10] in China. In terms of air entrainment characteristics, Chanson et al. [11] and Bung [12] analyzed the inception point of aeration and the sectional entrained air concentration of the step spillway, and proposed a calculation formula. Ghaderi et al. [13] compared the experiment and simulation results to verify the RNG k-ε turbulence model simulation analysis of the step spillway flow air entrainment characteristics. Raza et al. [14] studied the effect of slope on the initial air entrainment point, and they found that the steeper the slope, the shorter the length of the nonentrained flow zone. In terms of energy dissipation characteristics, the energy dissipation rate of the step spillway is much higher than that of the traditional smooth spillway, so the size of the downstream dissipation basin can be reduced [15]. Salmasi et al. [16] studied the effect of the step slope and number on the energy dissipation characteristics of the step spillway, and the results showed that the energy dissipation rate increased with the increase in the step slope number. Moreover, some researchers have proposed the use of a pool step-type spillway to increase the energy dissipation effect of the step spillway [17–19].

In summary, a lot of research has been conducted at the domestic and international levels on the flow pattern, pressure characteristics, air entrainment characteristics, and energy dissipation rate of the step-type energy dissipator. However, it has been found that the negative pressure near the convex corner of the vertical surface of the step is larger and easily able to produce cavitation damage, which when serious will affect the spillway flood dissipation effect, and even affect the safety of hydraulic buildings. Based on this, in this paper, we designed a combined trapezoidal energy dissipation baffle block-step energy dissipator; that is, the trapezoidal energy dissipation baffle block was arranged at the convex corner of the step. We used model tests and numerical simulation to study the effect of adding a trapezoidal energy dissipation baffle block on the air entrainment effect, pressure distribution, flow rate characteristics, and energy dissipation rate of the step energy dissipator. The main purpose was to reduce the negative step pressure, reduce the cavitation hazard, improve the energy dissipation rate, and provide a reference for the structural design of the trapezoidal energy dissipation baffle block-step combination energy dissipator.

## 2. Construction of Mathematical Models

The computational framework used by the Flow-3D software is a hybrid framework that dynamically tracks the free water surface using the Tru-VOF technique, which avoids the calculation of the gas phase and reduces the computational time [20]. Therefore, in this paper, Flow-3D software was used, and the RNG k-ε turbulence model was selected for numerical simulation. It has been shown that the RNG k-ε turbulence model can consider the effect of small-scale vortex motion and has better accuracy for the simulation of the step dissipator flow characteristics compared with the standard k-ε model [21,22]. The finite difference method was used to solve the algebraic equations iteratively; the VOF method was used to track the free water surface; the air entrainment model, density evaluation model, and drift-flux model were used to simulate the water–air two-phase flow.

### 2.1. Turbulence Model

The controlling equation for the RNGk-ε model is as follows:
Continuity equation:

$$\frac{\partial \rho}{\partial t} + \frac{\partial (\rho u_i)}{\partial x_i} = 0 \tag{1}$$

Momentum equation:

$$\frac{\partial(\rho u_i)}{\partial t} + \frac{\partial(\rho u_i u_j)}{\partial x_j} = -\frac{\partial \rho}{\partial x_i} + \frac{\partial}{\partial x_j}\left[(\mu + \mu_t)\left(\frac{\partial u_i}{\partial x_j} + \frac{\partial u_j}{\partial u_i}\right)\right] \tag{2}$$

$k$ equation:

$$\frac{\partial(\rho k)}{\partial t} + \frac{\partial(\rho k u_i)}{\partial x_i} = \frac{\partial}{\partial x_j}\left[\sigma_k(\mu + \mu_t)\frac{\partial k}{\partial x_j}\right] + G_k - \rho\varepsilon \tag{3}$$

$\varepsilon$ equation:

$$\frac{\partial(\rho\varepsilon)}{\partial t} + \frac{\partial(\rho\varepsilon u_i)}{\partial x_i} = \frac{\partial}{\partial x_j}\left[\sigma_\varepsilon(\mu + \mu_t)\frac{\partial\varepsilon}{\partial x_j}\right] + C_{1\varepsilon}\frac{\varepsilon}{k}G_k - C_{2\varepsilon}\rho\frac{\varepsilon^2}{k} \tag{4}$$

where $\rho$ is density; $t$ is time; $u_i$ and $u_j$ are velocity components; $x_i$ and $x_j$ are coordinate components; $\mu$ is the molecular viscosity coefficient; $\mu_t$ is the turbulent viscosity coefficient, taken as 0.0845; $\sigma_k$ and $\sigma_\varepsilon$ are the Prandtl numbers corresponding to the turbulent kinetic energy $k$ and the turbulent kinetic energy dissipation rate $\varepsilon$, respectively; $G_k$ is the turbulent kinetic energy generation term due to the average velocity gradient; $C_{1\varepsilon}$ and $C_{2\varepsilon}$ are constant terms, $C_{1\varepsilon} = 1.42$ and $C_{2\varepsilon} = 1.68$.

### 2.2. Air Entrainment Related Models

In this paper, we used the air entrainment model, the density evaluation model, and the drift-flux model to simulate the air entrainment in water. The air entrainment model was used to simulate the process of air entrainment at the free water surface. After the air was entrained into the water body, its diffusion and movement were controlled by the density evaluation and drift-flux models [23].

Air entrainment model: It was assumed that the air entrainment on the free surface of the water flow is controlled by the destabilizing force linearly related to the turbulent kinetic energy and the stabilizing force related to both surface tension and gravity. The air entrainment phenomenon occurs when the destabilizing force is larger than the stabilizing force.

$$L_T = \frac{\mu_T^{\frac{3}{4}}k}{\varepsilon} \tag{5}$$

$$P_t = \rho_m k \tag{6}$$

$$P_d = \rho_m g_n L_T + \frac{\sigma}{L_T} \tag{7}$$

$$S_a = \begin{cases} K_{air}A_s\left[\frac{2(P_t - P_d)}{\rho_m}\right] & P_t > P_d \\ 0 & P_t \leq P_d \end{cases} \tag{8}$$

where $L_T$ is the turbulence length scale; $\rho_m$ is the mixed-phase density; $g_n$ is the gravitational normal component to the water surface; $\sigma$ is the surface tension coefficient; $S_a$ is the volume of gas admixed into the mesh per unit time; $K_{air}$ is the scaling factor and the default value is 0.5.

Density evaluation model: After air entrainment occurs, the bubbles are transported and turbulently diffused by the water column, and the controlling equation is:

$$\frac{\partial c}{\partial t} + \frac{\partial}{\partial x_i}(U_{ai}c) - \frac{\partial}{\partial x_i}\left(D_c\frac{\partial c}{\partial x_i}\right) = \frac{S_a}{V_c} \tag{9}$$

where $c$ is the air admixture density; $U_{ai}$ is the velocity of motion of the air phase [10,24]; $D_c$ is the diffusion coefficient; $S_a$ is the air admixture source term in Equation (8); $V_c$ is

the mesh volume. The average density of the two-phase flow is calculated using the following equation:

$$\rho_b = (1 - c)\rho_w + c\rho_a \tag{10}$$

where $\rho_a$ is the air density; $\rho_b$ is the average density of the two phases of water and air; $\rho_w$ is the density of water.

Drift-flux model: This reflects the buoyancy force, interphase drag, and interaction between bubbles in the motion of the water–air two-phase flow. The model assumes that the slip velocity between the two phases of water and air is a constant, and the equation of motion of the air in water is:

$$\left(\frac{1}{\rho_w} - \frac{1}{\rho_a}\right)\nabla p = \left(\frac{(1 - c)\rho_w + c\rho_a}{c(1 - c)\rho_w\rho_a}\right)KU_r \tag{11}$$

where $U_r$ is the slip velocity, and $K$ is the interphase drag coefficient, calculated by the following equation:

$$K = \frac{\alpha}{2V_p}A_p\rho_c\left(C_dU_r + \frac{12\mu_c}{\rho_c R_p}\right) \tag{12}$$

where $A_p$ is the cross-sectional area of the bubble; $C_d$ is the custom resistance coefficient; $\rho_c$ is the density of the continuous phase; $\mu_c$ is the dynamic viscosity of the continuous phase; $V_p$ is the volume of a single bubble; $R_p$ is the bubble radius.

### 2.3. Meshing and Boundary Conditions

The mathematical model was divided into three parts: the diversion channel, the step stage, and the tailwater channel. In order to speed up the computational convergence, the mesh block was divided into three mesh blocks according to the diversion channel, the step stage, and the tailwater channel; a structured mesh was used, and the mesh refinement process was carried out for the mesh of the step stage, which was the significance study, with a grid size of 3 mm, as shown in Figure 1.

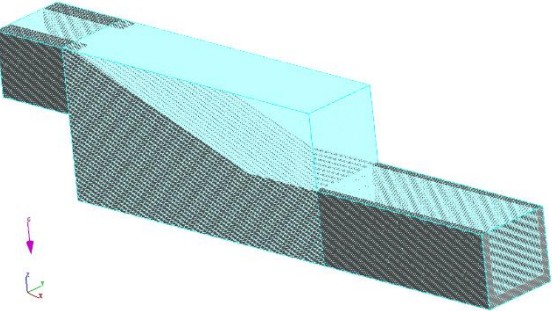

**Figure 1.** Meshing pattern in the computational domain.

### 2.4. Sensitivity Analysis of Grids

To determine the correctness of the mesh size, a mesh sensitivity analysis was performed for the step stage mesh using a square mesh of 8 mm, 6 mm, 4 mm, 3 mm, and 2 mm. In Figure 2, a comparison of the average water depth of the section at the 5th step convex angle position for the same flow condition is given for the five mesh sizes. From Figure 2, it can be seen that there are differences in the water flow section's water depth under different mesh sizes. The average water depth gradually decreases with the decrease in the mesh size, and gradually stabilizes when the mesh size is 3 mm and 2 mm. The maximum difference was only about 0.7%; a further reduction in the mesh size had little effect on improving the calculation accuracy. In addition, the computation time significantly increased when performing simulations with a mesh size of 2 mm, so a square mesh of 3 mm was used for this simulation.

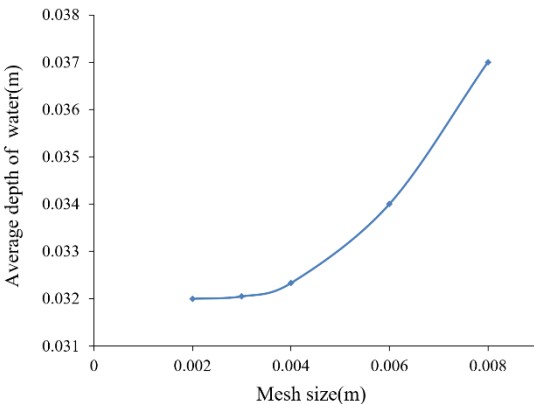

**Figure 2.** Mesh irrelevance test.

## 3. Experimental Validation of Mathematical Models

### 3.1. Experimental Systems

The experimental system in this paper mainly consisted of an upstream water tank, a diversion channel, a step stage, a tailwater channel, a water cistern, and an underground reservoir to form a circulatory system. A schematic diagram of the experimental device and model arrangement profile is shown in Figure 3. The length of the diversion channel was $L_1$ = 1 m and the width was W = 0.2 m. The step stage consisted of six steps, the slope ratio was 1:3, the length of a single step was $L_2$ = 0.18 m, the height was h = 0.06 m, and the width was s = 0.2 m. The model arrangement of two experimental schemes is shown in Figure 4: type I for the traditional step energy dissipator scheme; type II for the trapezoidal energy dissipation baffle block-step combination energy dissipator scheme. In the horizontal plane of the step at the convex angle position, the trapezoidal energy dissipation baffle block was placed. The trapezoidal energy dissipation baffle block lower bottom length was $l_{a1}$ = 0.02 m, the upper bottom length was $l_{a2}$ = 0.01 m, the width was $l_b$ = 0.02 m, and the height was $l_c$ = 0.02 m.

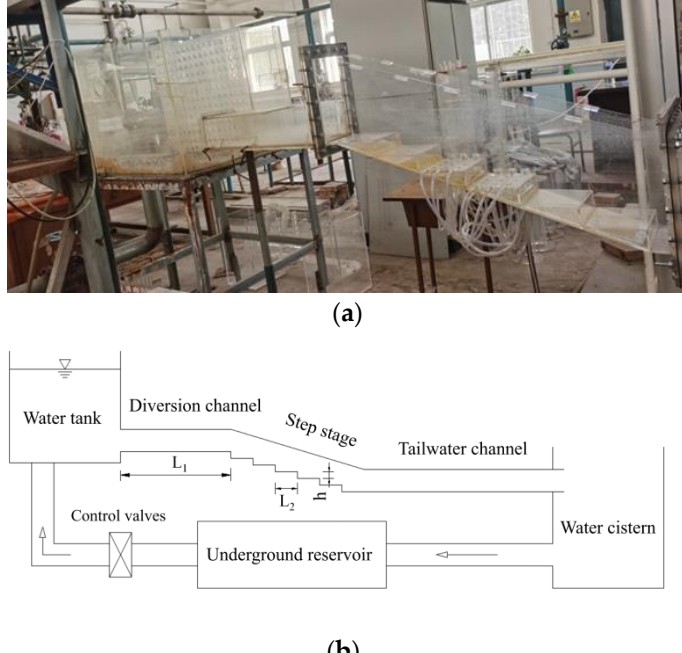

(**a**)

(**b**)

**Figure 3.** Model experimental setup and profile arrangement: (**a**) experimental model; (**b**) experimental model layout section.

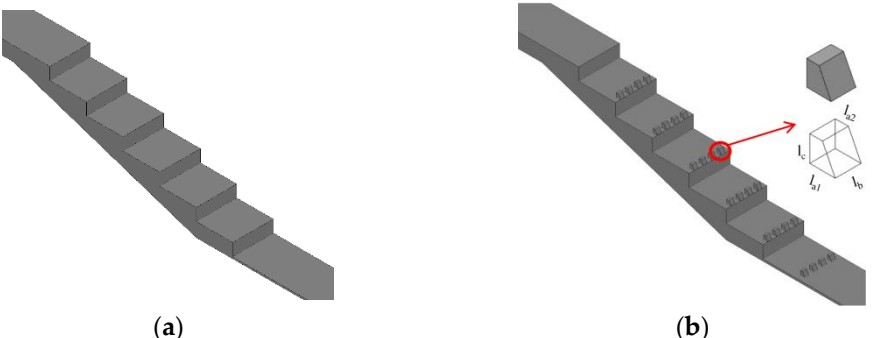

**Figure 4.** Schematic diagram of the experimental model arrangement: (**a**) Type I: traditional step energy dissipator; (**b**) Type II: trapezoidal energy dissipation baffle block-step combination energy dissipator.

The experiment used the dimensionless parameter $\zeta = h_k/h$ to characterize the flow rate, where $h_k$ is the critical water depth:

$$h_k = \sqrt[3]{\frac{q^2}{g}} \tag{13}$$

where $q$ is the water discharge per unit width, m$^2$/s; $g$ is the gravity acceleration constant, m/s$^2$. This paper selected four working conditions with $\zeta$ of 0.714, 0.936, 1.134, and 1.316 for the experiment.

### 3.2. Model Validation

The experimental and simulated values of the water surface lines of the two types of step water flow at $\zeta = 1.316$ were compared to ensure the reliability of the numerical simulation calculation, and the results are shown in Figure 5. We took the bottom plate downstream section 0-0 of the step as the reference plane. The vertical coordinate was the relative elevation head $d/d_c$, where $d$ is the elevation head of the water flow and $d_c$ is the total elevation of the step dissipator. The horizontal coordinate was position $x_1$ from the upstream of the step (see Figure 6 for the schematic diagram). As shown in Figure 5, the simulation calculation results were in good agreement with the experimental results. The maximum relative error along the water depth was 8.1%, proving that the simulation method was reasonable and the calculation results reliable.

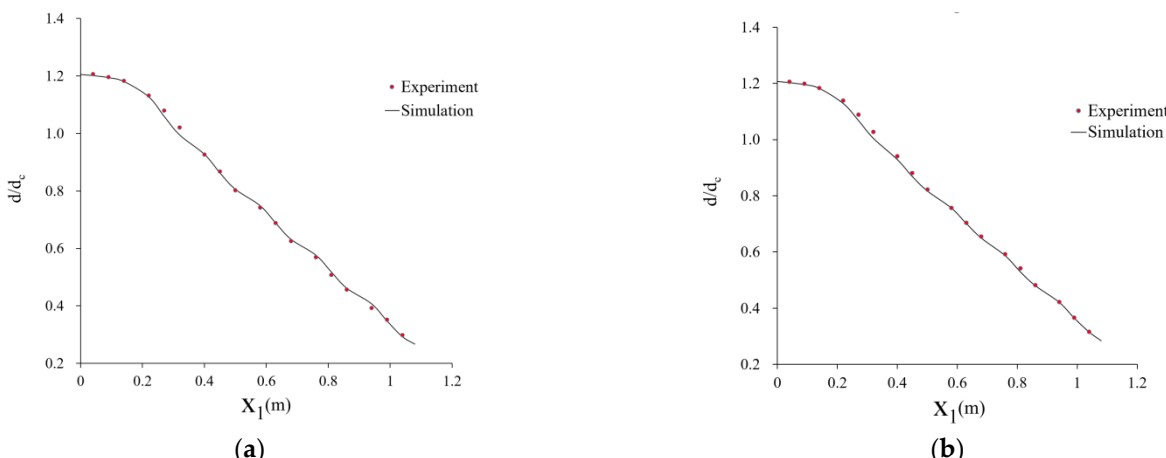

**Figure 5.** Comparison of simulated and experimental water depth: (**a**) Type I; (**b**) Type II.

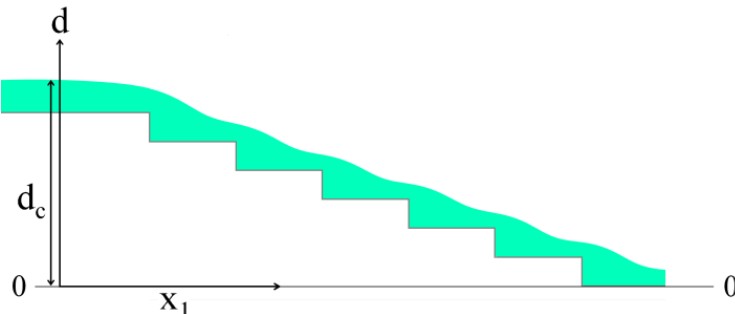

**Figure 6.** Water level diagram.

## 4. Results and Discussion

### 4.1. Air Entrainment Characteristics

Water flows on the steps under the effect of strong turbulence and the boundary layer development to the water surface. When the water surface is broken, the water quality points leap away from the surface of the water, and then, due to gravity, move back down, involving a large amount of air, such that the water is mixed with gas. The aerated flow can reduce the cavitation damage of the step dissipator. Therefore, it is crucial to study the air entrainment characteristics of the step dissipator.

Figure 7 compares the numerically simulated concentration of entrained air for type I and type II at different $\zeta$. The figure shows that the initial air entrainment point of both types of step energy dissipator moves down with the increase in the relative critical water depth. This is in conformity with the results obtained by [11]. Under each $\zeta$ working condition, the initial air entrainment point of type II is about one step ahead of type I; the entrained air concentration of type II is significantly larger than that of type I. Therefore, installing additional trapezoidal energy dissipation baffle blocks on the steps can reduce the length of the nonaeration flow region, increase the water-entrained air concentration, and reduce the cavitation damage risk of the steps.

The initial air entrainment point distance $L_c$ from the diversion channel was measured for both types of steps. Considering the steps as rough bodies, their roughness is expressed as $k_* = hcos\alpha$, and the friction coefficient f is expressed as:

$$f = \frac{q}{\sqrt{gk_*^3 sin\alpha}} \tag{14}$$

where $\alpha$ is the step slope and $q$ is the water discharge per unit width. The relationship between the step relative initial air entrainment point $L_c/h$ and f is shown in Figure 8. The figure shows that the initial air entrainment point of both type I and type II gradually moves backward with the increase in the flow rate. The initial air entrainment point of type II is more forward than type I. The difference $L_c/h$ between the initial air entrainment point of the two step energy dissipators gradually increases as the flow rate increases. Therefore, the trapezoidal energy dissipation baffle block-step combination energy dissipator, compared with the traditional step energy dissipator, can make the water flow in advanced aeration and slow down the increased flow so the initial air entrainment point location moves backward. The correlation equation between the initial air entrainment point location and the flow rate for two step-type dissipators was fitted using simulated data, as follows:

Type I:

$$\frac{L_c}{h} = 9.1556 f^{0.6482} \tag{15}$$

Type II:

$$\frac{L_c}{h} = 7.4226 f^{0.6981} \tag{16}$$

**Figure 7.** Step air entrainment concentration: (**a**) Type I, ζ = 0.714; (**b**) Type II, ζ = 0.714; (**c**) Type I, ζ = 0.936; (**d**) Type II, ζ = 0.936; (**e**) Type I, ζ = 1.134; (**f**) Type II, ζ = 1.134; (**g**) Type I, ζ = 1.316; (**h**) Type II, ζ = 1.316.

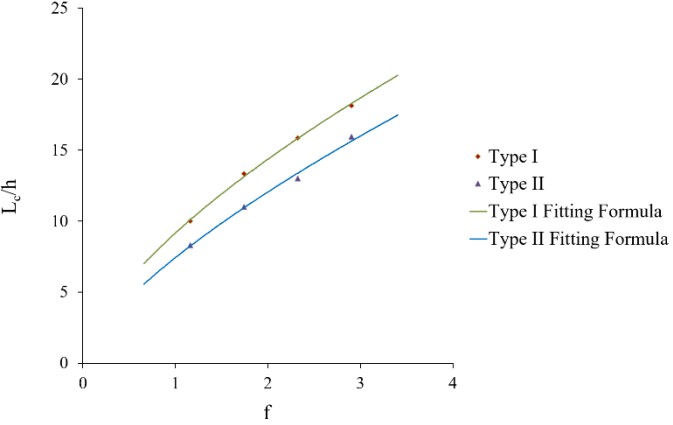

**Figure 8.** Step relative initial air entrainment point.

*4.2. Pressure Distribution*

Figure 9 shows the horizontal and vertical surface pressure distribution of both step types at $\zeta = 0.936$. In the legend, $x_2$ indicates the horizontal surface distance from the concave angle of the step, $z_1$ indicates the vertical surface distance from the concave angle of the step, and y indicates the different longitudinal profiles of the step. From Figure 9a, it can be seen that in the region of 0–0.6 $L_2$, the pressure variation pattern in the horizontal plane is basically the same, with an obvious "V" shape. This is in conformity with the results obtained by [5]. In the longitudinal section without the energy dissipation baffle block (y = 0.1 m), the maximum pressure values of type I and type II are the same. However, the minimum pressure value of type I is smaller than that of type II, and the position of the peak and trough of type I appears backward. For type II without the energy dissipation baffle block longitudinal section (y = 0.1 m) and with the energy dissipation baffle block longitudinal section (y = 0.075 m), the peak and trough appear at the same location. The minimum pressure value is the same, but the maximum pressure value is greater with the energy dissipation baffle block longitudinal section (y = 0.075 m). At 0.6–1 $L_2$, the pressure of type I decreases uniformly; the pressure of type II decreases slowly in the region of 0.6–0.8 $L_2$. After 0.8 $L_2$, the pressure of the longitudinal section without the energy dissipation baffle block (y = 0.1 m) decreases rapidly and generates negative pressure in the region of 0.93–1 $L_2$. In contrast, the longitudinal section with the energy dissipation baffle block (y = 0.075 m) makes the pressure increase from 0.8 $L_2$ to the headwater surface of the energy dissipation baffle block, due to the obstruction of the water flow by the energy dissipation baffle block. From the cloud diagram of the step pressure distribution (Figure 10), it can be seen that by arranging the energy dissipating baffle blocks at the convex corner of the step, the pressure distribution on the horizontal surface of the step can be made more uniform. The impact damage of water flow on the horizontal surface of the step can be reduced.

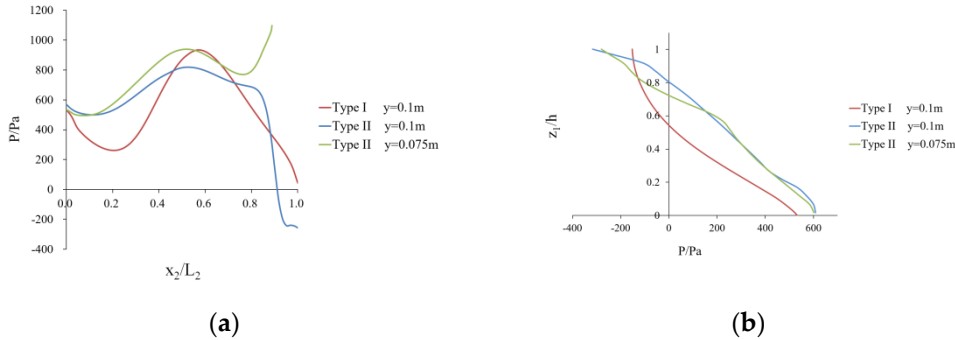

(a)          (b)

**Figure 9.** Pressure distribution on the horizontal and vertical surfaces of the step: (**a**) horizontal surface pressure distribution graph; (**b**) vertical surface pressure distribution diagram.

From Figure 9b, it can be seen that the pressure of both types tends to decrease on the vertical surface of the step. This is in conformity with the results obtained by [5]. In the interval of 0–0.8 h, the pressure of type II is greater than that of type I; in the interval of 0.9–1 h, the pressure of type II is less than that of type I. The negative pressure range of type II is smaller than that of type I, but the type II negative pressure maximum is larger than that of type I. The pressure intensity of the longitudinal section of type II without the energy dissipation baffle block (y = 0.1 m) and with the energy dissipation baffle block (y = 0.075 m) is basically the same in the 0–0.6 h interval. After 0.6 h, the pressure is higher in the longitudinal section of the nondissipated baffle block (y = 0.1 m), but the same pressure value is found in the step convex angle position. As can be seen from the cloud diagram of the step distribution (Figure 10), the arrangement of the energy dissipation baffle block at the convex corner of the step can increase the pressure on the vertical surface of the step and reduce the negative pressure area, thus reducing the cavitation area on the vertical surface of the step. However, a larger negative pressure will be generated near

the convex corner of the step, and corresponding measures need to be taken to prevent cavitation damage in the project.

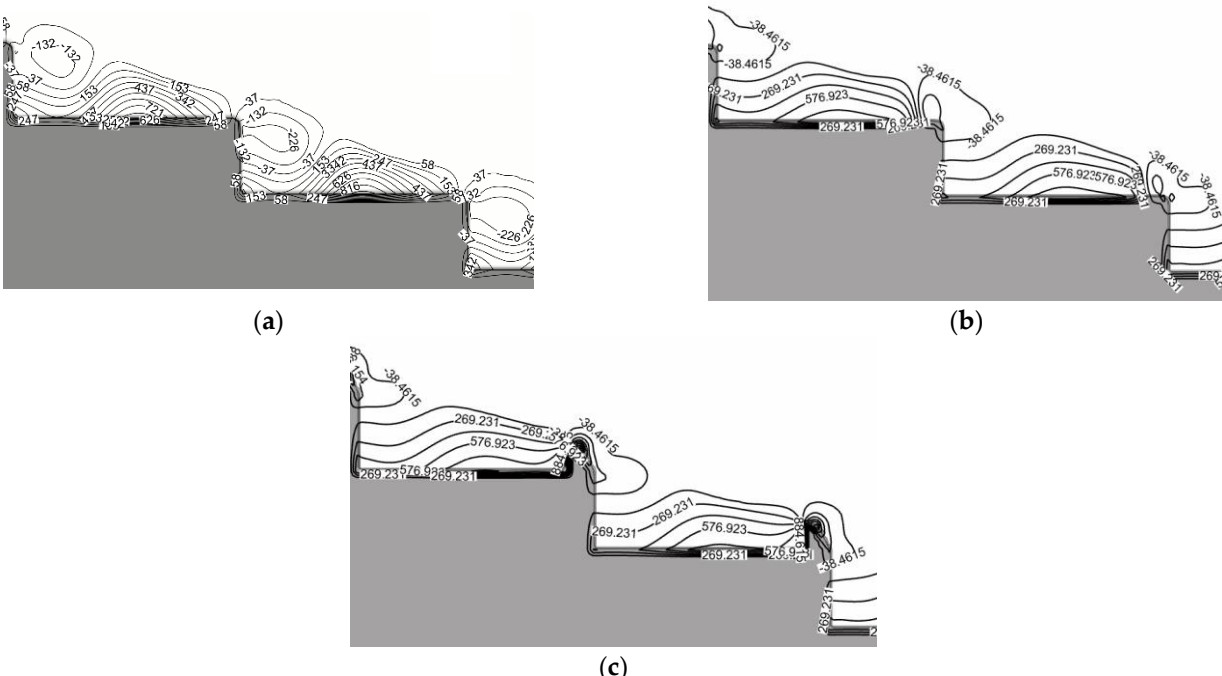

(**a**)          (**b**)

(**c**)

**Figure 10.** Cloud diagram of the pressure distribution on the step surface of different types: (**a**) Type I, y = 0.1 m; (**b**) Type II, y = 0.1 m; (**c**) Type II, y = 0.075 m.

The pressure change near the trapezoidal energy dissipation baffle block is shown in Figure 11. Due to the obstructing effect of the trapezoidal energy dissipation baffle block on the water flow in the mainstream area, the trapezoidal energy dissipation baffle block produces a sizeable positive pressure on the headwater surface. The top and backwater surfaces are prone to negative pressure, and the backwater surface is prone to the phenomenon of water flowing off the wall at low flow rates. Therefore, trapezoidal energy dissipation baffle blocks reduce the impact and cavitation damage to the steps. At the same time, they will be subject to cavitation damage. However, compared to the repair costs after impact and cavitation damage to the steps, replacing trapezoidal energy dissipation baffle blocks is less costly and more efficient.

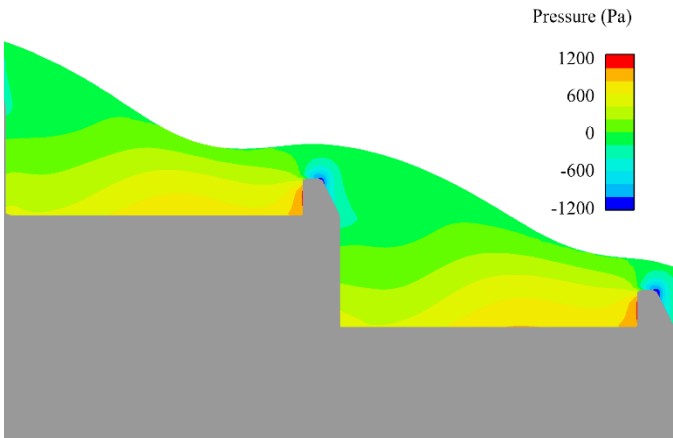

**Figure 11.** Cloud diagram of the pressure distribution in the type II section at ζ = 0.936.

### 4.3. Flow Velocity Distribution

Figure 12 shows the section flow velocity distribution at the convex angle position for different types of steps at $\zeta = 0.936$, where $z_2$ is the water depth perpendicular to the virtual bottom plate, $z_{max}$ is the maximum water depth of the section, V is the section flow velocity, and $V_c$ is the critical flow velocity. It can be seen that the flow velocity of the two types of step dissipators tends to increase gradually from the convex angle of the step to the water surface and decreases near the water surface position. This is in conformity with the results obtained by [19]. The flow velocity variation of type I from the convex corner of the step to the water surface is slight. The flow velocity at all water depths is greater than that of type II, so type II is more fully dissipating energy. Type II in the longitudinal section with the trapezoidal energy dissipation baffle block (y = 0.075 m) flow velocity variation is largest. In the longitudinal section without the trapezoidal energy dissipation baffle block (y = 0.1 m), the flow velocity variation is smaller; in 0.2–0.4, the water depth appears to be an abnormal region of higher flow velocity. The flow around the blunt-body principle suggests that the trapezoidal energy dissipation baffle block blocks the flow in the mainstream area, so the energy dissipation baffle block side of the water flow velocity increases, resulting in an abnormally larger flow velocity area. The backwater surface of the energy dissipation baffle block produces backflow and the flow velocity is lower, which makes the section flow velocity vary in a wide range.

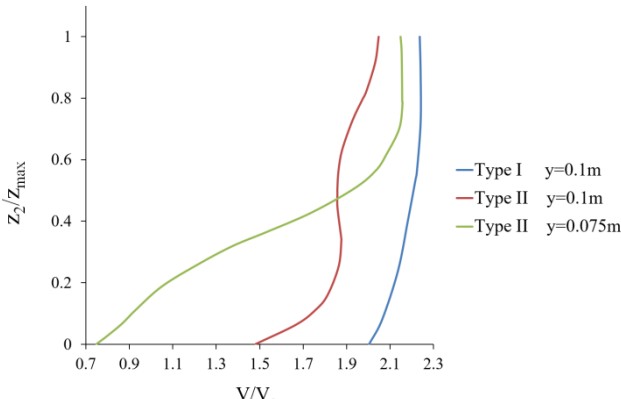

**Figure 12.** Section flow velocity distribution of different types of steps.

Figure 13 shows the longitudinal section flow velocity vector and cloud diagram for different types of steps at $\zeta = 0.936$. It can be seen that the concave angle of the steps forms an extensive range of vortex areas, and the vortex reduces the energy of the mainstream by producing strong turbulent shear and momentum exchange with the upper mainstream, achieving energy dissipation. The size of the vortex area of the two types of step energy dissipators in the figure is basically the same. However, the range of low flow velocity (V < 0.5 m/s) in the vortex area of type II is more extensive than that of type I, and the flow velocity is lower, so the energy dissipation effect is better.

To further study the vortex structure of the trapezoidal energy dissipation baffle block-step combination energy dissipator, this paper used the *Q*-criterion [25] for vortex identification. The *Q*-criterion approach is based on the characteristic equation of the gradient tensor and identifies the region where the second matrix invariant *Q* > 0 as a vortex. *Q* is defined as follows:

$$Q = \frac{1}{2}\left(\|B\|_F^2 - \|A\|_F^2\right) \tag{17}$$

where *A* is the symmetric part of the velocity gradient tensor, corresponding to the deformation in the flow field; *B* is the anti-symmetric part, corresponding to the rotation in the flow field. The symmetric tensor A has the effect of counteracting the rotation of the anti-symmetric tensor B rigid body, so the physical meaning of the *Q*-criterion is that the

vorticity of the rotational motion in the flow field is greater than the deformation motion, which dominates.

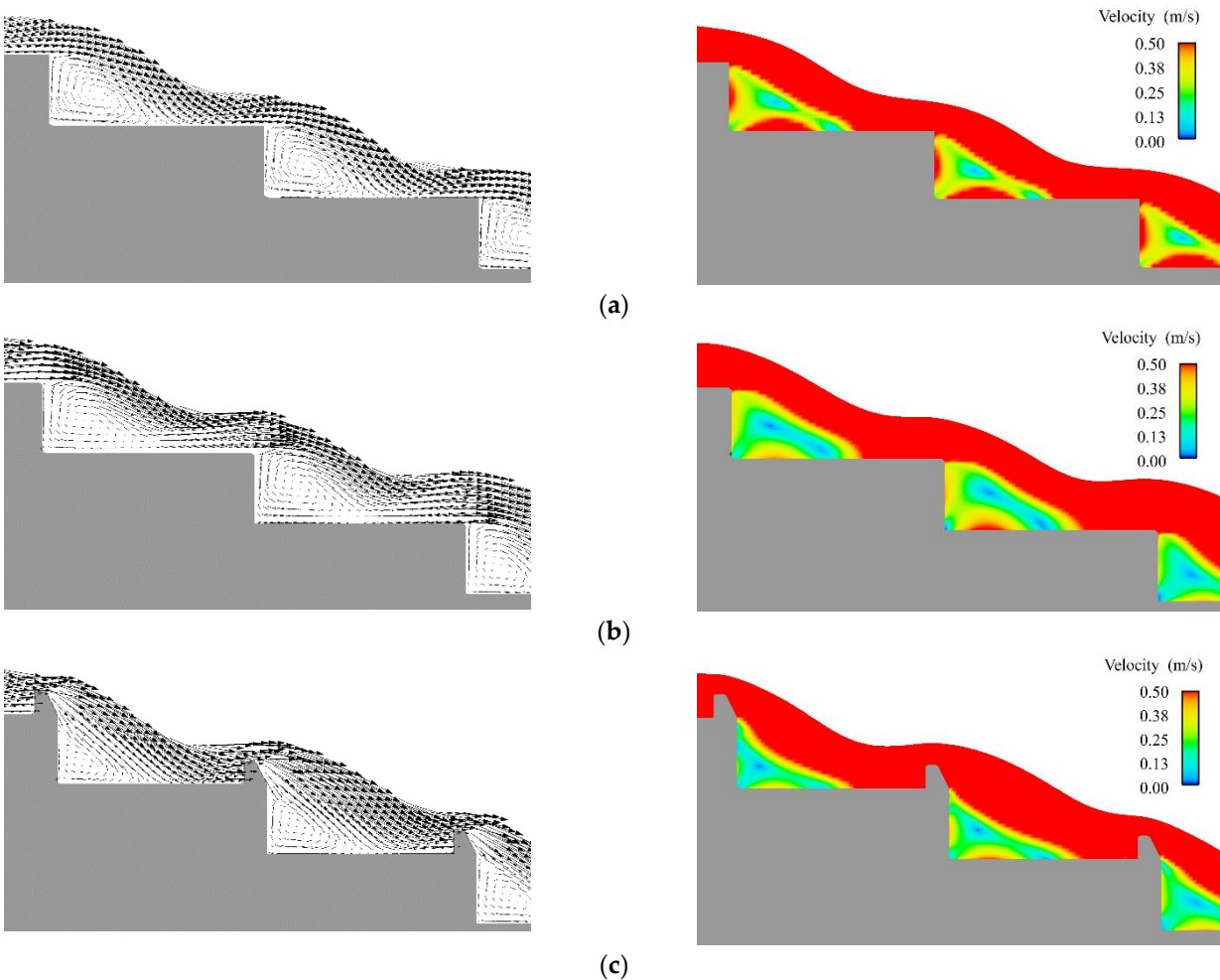

**Figure 13.** Flow velocity vector and cloud diagrams of longitudinal sections for different types of steps: (**a**) Type I, y = 0.1 m; (**b**) Type II, y = 0.1 m; (**c**) Type II, y = 0.075 m.

Figure 14 shows the *Q*-equivalent surface of the two types of step dissipators for ζ = 1.134. It can be seen that a large number of vortex structures exist near the convex angle of the step dissipator of type I. This is due to the sudden change of step boundary, which causes the separation of the water flow boundary layer and the formation of the return flow spin-roll zone and the upper mainstream zone on the step. The junction area between the mainstream zone and the spin-roll zone produces a strong shear layer because of the large flow velocity gradient, thus forming a vortex structure of greater strength [26]. Furthermore, due to the development of boundary layers, a small number of vortex structures are present near the sidewalls on both sides of the steps. The vortex structure near the convex corner of the step dissipator of type II consists of two parts. One is the same as type I, formed by the strong shear layer between the mainstream and spin-roll zones. The second is the vortex structure formed around the dissipation baffle block due to the flow around bluff bodies generated when the water passes through the trapezoidal dissipation baffle block [27]. Therefore, the trapezoidal energy dissipation baffle block is arranged at the convex corner position of the step, which makes the distribution of the step vortex structure change greatly, and helps the energy dissipation of the step energy dissipator.

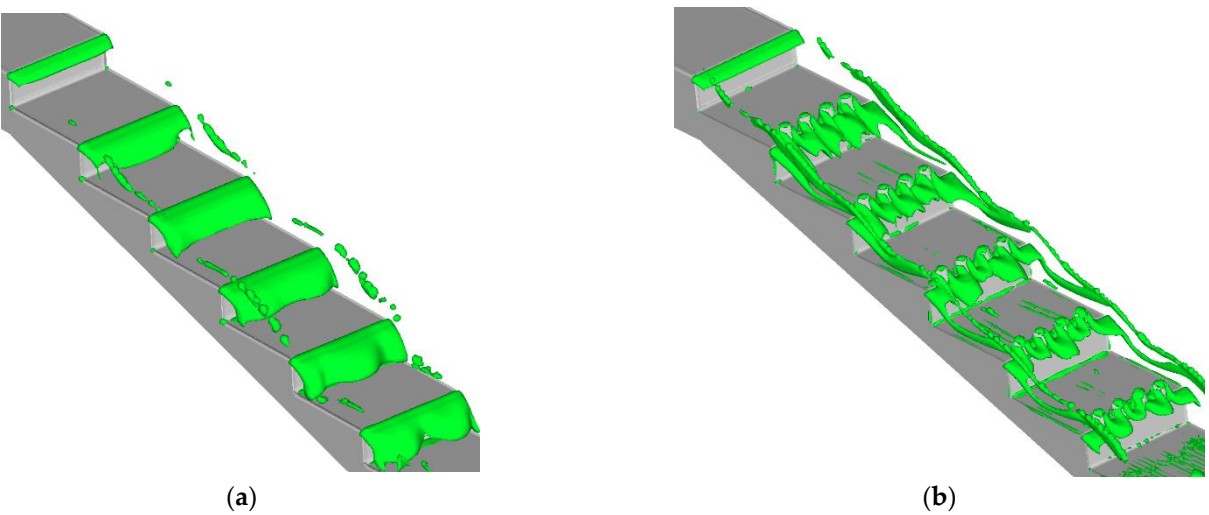

(**a**)　　　　　　　　　　　　　　　　　　　　(**b**)

**Figure 14.** The Q-equivalent surface of two types of step energy dissipators: (**a**) Type I; (**b**) Type II.

### 4.4. Energy Dissipation Rate Analysis

In order to compare and analyze the energy dissipation effect of two types of steps, two sections near the end of the upstream diversion channel and the downstream of the last step were selected, and the energy dissipation rate was calculated using the following formula with the downstream bottom plate as the reference surface:

$$\eta = \frac{E_0 - E_1}{E_0} \times 100\% \tag{18}$$

where $E_0$ is the total upstream energy and $E_1$ is the total downstream energy. The calculation results are shown in Figure 15.

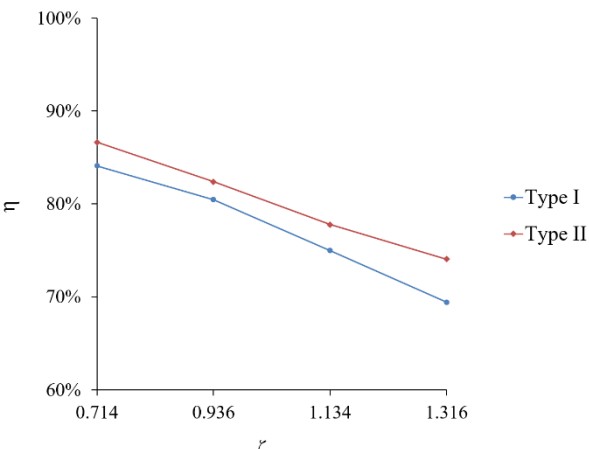

**Figure 15.** Energy dissipation rate of two types of step dissipators.

The results showed that within the experimental range, the energy dissipation rate of both types of step energy dissipators was above 70%. Under each flow condition, the energy dissipation rate of type II was significantly larger than that of type I, and the maximum increase in its energy dissipation rate was up to 6.67%. The energy dissipation rate of both types of step energy dissipators decreased with the increase in flow, but the decreasing energy dissipation rate of type II was lower than that of type I. Therefore, arranging trapezoidal energy dissipation baffle blocks at the convex corners of the steps can enhance the effect of step energy dissipation, and reduce the scouring damage of water flow downstream.

## 5. Conclusions

(1)  Compared with the traditional step energy dissipator, the trapezoidal energy dissipation baffle block-step combination energy dissipator can increase the energy dissipation rate by about 6.67%, with better energy dissipation characteristics. The use of a trapezoidal energy dissipation baffle block to improve the hydraulic characteristics of the step energy dissipator is feasible.

(2)  The trapezoidal energy dissipation baffle block-step combination energy dissipator has a better air entrainment effect. Compared with the traditional step energy dissipator, the trapezoidal energy dissipation baffle block-step combination energy dissipator can advance the initial aeration point by one step and increase the air entrainment volume of the flow. We also proposed the calculation formula of the initial aeration point of the trapezoidal energy dissipation baffle block-step combination energy dissipator.

(3)  The trapezoidal energy dissipation baffle block-step combination energy dissipator has a lower risk of cavitation. By adding trapezoidal energy dissipation baffle blocks at the convex corner of the step, the pressure variation law of the horizontal and vertical surfaces of the traditional step is changed, which reduces the extreme value of the pressure on the horizontal surface of the step and reduces the distribution area of the negative pressure on the vertical surface of the step, thus reducing the risk of cavitation of the step. The top and the backwater surface of the trapezoidal energy dissipation baffle block have negative pressure, which can easily cause cavitation damage, and certain protective measures will be needed for the actual project.

(4)  The trapezoidal energy dissipation baffle block-step combination energy dissipator has a lower flow velocity. The trapezoidal energy dissipation baffle block-step combination energy dissipator mainstream cross-sectional flow velocity still follows the law of a small bottom layer and large surface layer. However, compared with the traditional step energy dissipator, the trapezoidal energy dissipation baffle block-step combination energy dissipator mainstream section flow velocity change amplitude is higher, the flow velocity is lower, the concave angle roll area low-flow-velocity range is higher, and the energy dissipation effect is better. The trapezoidal energy dissipation baffle block-step combination energy dissipator makes the vortex structure more distributed in the vicinity of the trapezoidal energy dissipation baffle block, which helps the energy dissipation of the step energy dissipator.

**Author Contributions:** Data curation, Y.T. and Y.L.; funding acquisition, X.S.; investigation, Y.T. and Y.L.; writing—original draft, Y.T.; writing—review and editing, X.S. and Y.L. All authors have read and agreed to the published version of the manuscript.

**Funding:** The research was funded by the National Natural Science Foundation of China (51179116).

**Institutional Review Board Statement:** Not applicable.

**Informed Consent Statement:** Not applicable.

**Data Availability Statement:** Some or all of the data, models, or code that support the findings of this study are available from the corresponding author upon reasonable request.

**Acknowledgments:** This research was supported by the Collaborative Innovation Center of New Technology of Water-Saving and Secure and Efficient Operation of Long-Distance Water Transfer Project at the Taiyuan University of Technology.

**Conflicts of Interest:** The authors declare no conflict of interest. The funders had no role in the design of the study; in the collection, analyses, or interpretation of data; in the writing of the manuscript; or in the decision to publish the results.

**Nomenclature**

| | |
|---|---|
| $A_p$ | the cross-sectional area of the bubble (m$^2$) |
| $C_{1\varepsilon}$ | constant terms (-) |
| $C_{2\varepsilon}$ | constant terms (-) |
| $C_d$ | the custom resistance coefficient (-) |
| $c$ | the air admixture density (-) |
| $D_c$ | the diffusion coefficient (-) |
| $d$ | the elevation head of the water flow (m) |
| $d_c$ | the total elevation of the step dissipator (m) |
| $f$ | friction coefficient (-) |
| $G_k$ | the turbulent kinetic energy generation term (kg/m/s$^{-3}$) |
| $g_n$ | the gravitational normal component to the water surface (m/s$^2$) |
| $h$ | step height (m) |
| $h_k$ | critical water depth (m) |
| $K$ | interphase resistance coefficient (-) |
| $K_{air}$ | scale factor (-) |
| $K*$ | roughness (m) |
| $L_1$ | length of diversion channel (m) |
| $L_2$ | step length (m) |
| $L_c$ | initial air entrainment point distance (m) |
| $L_T$ | turbulence length scale (m) |
| $l_{a1}$ | energy dissipating baffle block bottom length (m) |
| $l_{a2}$ | energy dissipating baffle block upper length (m) |
| $l_b$ | energy dissipating baffle block width (m) |
| $l_c$ | energy dissipating baffle block height (m) |
| $q$ | single wide flow (m$^3$/s) |
| $R_p$ | air bubble radius (m) |
| $S_a$ | volume of gas blended into the grid per unit time (-) |
| $t$ | time (s) |
| $U_{ai}$ | velocity of motion of gas phase (m/s) |
| $U_r$ | slip speed (m/s) |
| $u_i$ | i-direction velocity component (m/s) |
| $u_j$ | j-direction velocity component (m/s) |
| $V$ | cross-sectional flow rate (m/s) |
| $V_b$ | grid volume (-) |
| $V_c$ | critical flow rate (m/s) |
| $V_p$ | volume of a single bubble (-) |
| $W$ | width of diversion channel (m) |
| $x_1$ | location upstream from the step (m) |
| $x_2$ | distance from the horizontal plane of the concave angle of the step (m) |
| $x_i$ | i-directional coordinate components (-) |
| $x_j$ | j-directional coordinate components (-) |
| $y$ | steps in different longitudinal sections (m) |
| $z_1$ | distance from the vertical plane of the concave angle of the step (m) |
| $z_2$ | water depth perpendicular to the virtual substrate (m) |
| $z_{max}$ | maximum water depth at section (m) |
| $\alpha$ | slope (-) |
| $\zeta$ | dimensionless parameters (-) |
| $\eta$ | energy dissipation (-) |
| $\mu$ | molecular viscosity coefficient (-) |
| $\mu_c$ | power viscosity of continuous phase (N·s/m$^2$) |
| $\mu_t$ | turbulent viscosity coefficient (-) |
| $\rho$ | density (kg/m$^3$) |
| $\rho_a$ | density of air (-) |

| $\rho_b$ | average density of water and gas phases (kg/m$^3$) |
| $\rho_c$ | density of continuous phase (-) |
| $\rho_m$ | mixed-phase density (kg/m$^3$) |
| $\rho_w$ | density of water (kg/m$^3$) |
| $\sigma$ | surface tension coefficient (-) |
| $\sigma_k$ | the Prandtl number corresponding to the k (-) |
| $\sigma_\varepsilon$ | the Prandtl number corresponding to the $\varepsilon$ (-) |

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
