# Peer review of "Study on the Hydraulic Characteristics of the Trapezoidal Energy Dissipation Baffle Block-Step Combination Energy Dissipator"

_water, doi:10.3390/w14142239_

Round 1
Reviewer 1 Report
The authors undertook an extensive experimental campaign and numerical simulation to study hydraulic characteristics. It uses a combination of hydraulic model experiments and numerical simulation to study hydraulic characteristics of step-type energy dissipator. They have added a trapezoidal energy dissipation pier at the convex corner of the traditional step to form a trapezoidal energy dissipation pier - step combination. This is to improve the air entrainment characteristics, energy dissipation and to reduce the influence of cavitation. Overall, the article has novelty with acceptable methods and it is well drafted. Minor corrections
Why authors didn't proceed to the variation of the dimensions (a), (b) and (c) of the work to study the impact of the size of the trapezoidal energy dissipation pier?
I suggest to justify the selection of these dimensions length a = 0.02m, width b = 0.02m, height c = 0.02m).
Reviewer 2 Report
There are some interesting results here which other researchers could benefit from.
The overall thrust of the paper in terms of the work done and the use of physical and mathematical models to look at the proposed layout is all okay. However there are a few issues that need addressing before publication can proceed.
Firstly the language is not appropriate for an international audience, and requires significant editing for clarity. Before reviewing any revision of this work I would expect this to be passed to a professional editing service for editing by a subject specialist. Without this, the impact of this work will be minimal. It is important to stress in cases like these that it is not simply a case of correcting the grammar and writing style. There is also a need to change the text to make it clearer what is going on, by presenting a logical sequence of ideas in terms readers can understand. An experienced author or editor can help with this.
Some of the terminology is not that clear. For instance, I think the word 'pier' is not as clear as the alternative 'block' or 'baffle block' and so I suggest this word is used instead. Alternatively please make it clear that there are different words that can be used to mean the same thing.
Many different symbols (d, h etc.) are used with different subscripts. A glossary of terms would be helpful.
Aims of the work (l.63-73) - these could be improved to explain what experiments are being conducted and what the key objectives are in each case. It is also not clear what the motivation is in terms of engineering design and practice in China or elsewhere in the world. Baffle blocks (piers) are a well known component of many spillways. In what way is this work novel, and in what ways does it confirm what we already know? Furthermore, is cavitation damage in spillways actually a problem that needs solving?
Some of the experimental design is not well explained or justified. Why this combination of L and W for example, why this particular layout of blocks? W does not seem large enough to me to ensure representativeness of a real spillway, standard width for such experiments should be at least 0.3m in my view to minimise the effects of wall drag. Another query is, what is the fourth dimension on the block, i.e., the length of the side parallel to a? This is not given. Furthermore, there is no clear explanation of what is actually being measured. Water levels and position of entrainment point? Please clarify for the benefit of readers.
I think more could be done in the Discussion and Conclusions to refer the reader back to the results of some of the earlier studies by Chanson and others
Citations and referencing, more broadly, for an international journal it is not acceptable to have half the references taken from Chinese publications as it is at present. There is a wealth of recent literature on step-spillways and modelling in international journals which should be cited. This would help readers by referring to a wider set of perspectives.
Round 2
Reviewer 2 Report
Having scrutinised the latest version of Water article 1737319 together with the authors' responses I am satisfied this is ready for publication now